# Multi-Chemical Omics Analysis of the Symbiodiniaceae *Durusdinium trenchii* under Heat Stress

**DOI:** 10.3390/microorganisms12020317

**Published:** 2024-02-02

**Authors:** Jennifer L. Matthews, Maiken Ueland, Natasha Bartels, Caitlin A. Lawson, Thomas E. Lockwood, Yida Wu, Emma F. Camp

**Affiliations:** 1Climate Change Cluster, University of Technology Sydney, Ultimo, NSW 2007, Australia; 2Centre for Forensic Sciences, School of Mathematical and Physical Sciences, University of Technology Sydney, Ultimo, NSW 2007, Australia; 3Hyphenated Mass Spectrometry Laboratory, School of Mathematical and Physical Sciences, University of Technology Sydney, Ultimo, NSW 2007, Australia; 4School of Environmental and Life Sciences, University of Newcastle, Ourimbah, NSW 2258, Australia

**Keywords:** chemical analysis, climate change, conservation, elementomics, metabolomics, volatilomics

## Abstract

The urgency of responding to climate change for corals necessitates the exploration of innovative methods to swiftly enhance our understanding of crucial processes. In this study, we employ an integrated chemical omics approach, combining elementomics, metabolomics, and volatilomics methodologies to unravel the biochemical pathways associated with the thermal response of the coral symbiont, Symbiodiniaceae *Durusdinium trenchii*. We outline the complimentary sampling approaches and discuss the standardised data corrections used to allow data integration and comparability. Our findings highlight the efficacy of individual methods in discerning differences in the biochemical response of *D. trenchii* under both control and stress-inducing temperatures. However, a deeper insight emerges when these methods are integrated, offering a more comprehensive understanding, particularly regarding oxidative stress pathways. Employing correlation network analysis enhanced the interpretation of volatile data, shedding light on the potential metabolic origins of volatiles with undescribed functions and presenting promising candidates for further exploration. Elementomics proves to be less straightforward to integrate, likely due to no net change in elements but rather elements being repurposed across compounds. The independent and integrated data from this study informs future omic profiling studies and recommends candidates for targeted research beyond Symbiodiniaceae biology. This study highlights the pivotal role of omic integration in advancing our knowledge, addressing critical gaps, and guiding future research directions in the context of climate change and coral reef preservation.

## 1. Introduction

The Anthropocene is characterised by unprecedented human-driven change in both land and marine ecosystems [1]. Coral reefs are an ecosystem that globally is under threat from the rate and extent of environmental change occurring, with 14% of the world’s coral reefs lost between 2009 and 2018 due to elevated sea surface temperatures alone [2]. Such drastic loss to an iconic ecosystem that supports socio-ecological systems globally has created an urgent need for a new understanding of the mechanisms that support coral resilience, particularly to climate-related stressors (e.g., temperature) [3,4,5]. As corals are holobionts, the coral animal, their associated symbiotic algae Symbiodiniaceae, and other microorganisms (e.g., bacteria, viruses, fungi, etc. [6]) all contribute to the coral’s observed phenotype during stress [7,8]. When coral symbiotic partners are studied in hospite, deconvolving their role relative to other holobiont partners can be challenging. Consequently, researchers have turned to culturing symbiotic partners, such as Symbiodiniaceae, ex hospite to further our understanding of their functional diversity and stress response (e.g., [9,10,11]).

Symbiodiniaceae are critical to the health of the coral host, translocating carbon, providing metabolites, and bioaccumulating excess trace metals to reduce toxicity [12,13,14,15,16]. Changes in coral health during thermal stress have been correlated with their type of Symbiodiniaceae [17,18,19]. However, complex site-specific partnerships have also been documented (e.g., [20]), demonstrating the role local environmental conditions play in shaping partnerships. When studied ex hospite, phylogenetic differences have explained the stress response of cultured Symbiodiniaceae (e.g., [21]); however, they alone do not often account for the diversity of observed characteristics (traits). Consequently, “functional types” have been used as a unit to determine the stress response of Symbiodiniaceae and ultimately predict their ecological success [22]. To date, functional types have typically been based on photosynthetic functioning (e.g., photosystem II (PSII) maximum photochemical efficiency, *F*_v_*/F*_m_; [22]), and more recent work has also identified elemental types characteristic of phylogeny and environment [9,23,24]. To understand the complexity of the organism’s response to stress, however, requires a more complete understanding of the organism that moves beyond discrete traits and single-method analyses [25,26].

Genomic studies of Symbiodiniaceae currently dominate the field, predominantly in an effort to characterise the immense genetic and functional diversity of Symbiodiniaceae and assist in the identification of Symbiodiniaceae genotypes and phenotypic interpretations during experiments [27]. With an increase in genomic information [28,29,30,31,32,33] and the accessibility of high-throughput sequencing, it is no surprise that gene expression studies (e.g., transcriptomics) have increased, revealing key phenotypic traits, genes, and molecular mechanisms of interest for future study [34,35,36]. Transcriptomics informs us of gene expression patterns but cannot reveal protein end-products or the numerous post-transcriptional modifications to RNA (e.g., RNA methylation, spliced transcripts) that can further influence gene transcription. Meanwhile, elementomics, metabolomics, and volatilomics, despite their promise as highly informative tools [37], are still relatively new applications in coral and Symbiodiniaceae biology, where costs, access to methodology and analytical equipment, and a lack of reference elementomes, metabolomes, and unknown compounds continue to impede their application. However, when different ‘omics techniques are integrated, e.g., transcriptomics with metabolomics [16], proteomics with metabolomics [38], and microbiomics and metabolomics [39], a clearer understanding of the physiological responses can be achieved. Multi-omic analyses have been applied to in hospite Symbiodiniaceae; for example, the integration of transcriptomics with metabolomics has helped reveal Symbiodiniaceae diversification and interactions with host cnidarians at the functional level [16]. However, there are still remarkably few integrated multi-omic studies of Symbiodiniaceae, either in hospite or in culture. Proteome, metabolome, and transcriptome data for three Symbiodiniaceae cultures under control and heat stress exist [26], but challenges in data integration have limited the systems biology approach being applied. Nevertheless, as advances in technology occur, obtaining diverse omics datasets becomes more achievable, and thus resolving how to integrate diverse omics datasets is crucial for advancing our understanding of Symbiodiniaceae both in hospite and ex hospite [27,37].

In this paper, we collect elementome (the suite of elements making up the individual), metabolite (the substrates, intermediates, and end products of cellular metabolism), and volatile (biogenic volatile organic compounds released by the organism) data for Symbiodiniaceae, *Durusdinium trenchii* at two temperatures (26 °C and 33 °C) to assess how a chemical multi-omics approach can advance our understanding of the function of cultured Symbiodiniaceae under heat stress.

## 2. Materials and Methods

### 2.1. Culturing and Experimental Conditions

Symbiodiniaceae cultures of *D. trenchii* (ITS2: D1a, culture ID: SCF082) were subcultured (*n* = 6) from existing stocks at the University of Technology Sydney by adding 10 mL of original cultures in 90 mL of autoclaved and filter-sterilised (0.22 µm) artificial seawater (ASW) as per [40] and F/2 media. Cultures were grown for two months (to achieve a minimum cell density of 10^6^ cells/mL) at 26.0 °C with an irradiance of 85 ± 15 µmol photons m^−2^ s^−1^ (Philips TLD 18W/54 fluorescent tubes, 10,000 K on a 12 h:12 h light:dark cycle). Before use, cells were centrifuged at 700× *g* for 10 min at 26 °C and rinsed twice with ASW to remove residual media solution. Cells were resuspended in 100 mL ASW + F/2 media in sterile culture flasks.

Cultures were moved to control and heat stress water baths (*n* = 2 per treatment, similar to [26], both set initially to 26.0 ± 0.5 °C, with an irradiance of 107 ± 0.05 µmol photons m^−2^ s^−1^ (Philips TLD 18W/54 fluorescent tubes, 10,000 K on a 12 h:12 h light:dark cycle). The heat-stress water baths had the temperature increase by 1 °C day^−1^ for eight consecutive days until reaching approximately 33 °C on day nine. Control tanks remained at 26 °C ± 0.01 over the experimental period.

### 2.2. Photophysiology

Photophysiological performance of Symbiodiniaceae was assessed using a Soliense LIFT (Light Induced Fluorescence Transient)-FRR (Fast Repetition Rate) fluorometer (LIFT-FRRf; Soliense Inc., Shoreham, NY, USA) [41]. A 1 mL aliquot of each culture was collected directly from culture flasks in a biosafety cabinet daily at the same time each day, at the start of the light cycle. Samples were low-light (ca. 5–10 µmol photons m^−2^ s^−1^) acclimated for at least 30 min prior to measurements. An aliquot of 100 µL for each culture replicate (total *n* = 12, consisting of 6 replicates for each of the control and heat stress treatments) was transferred to the LIFT-FRRf optical chamber and diluted with 900 µL ASW. Excitation was delivered using a blue LED excitation source (peak excitation 470 nm), delivering single turnover fluorescent transients of 100 flashlets of 1.6 µs at 2.5 µs, followed by 127 flashlets of 1.6 µs, using a customised protocol to evaluate low light-acclimated photophysiology (10 µmol photons m^−2^ s^−1^ for 20 s), and light-acclimated photophysiology above growth light intensity (150 µmol photons m^−2^ s^−1^ for 60 s). All fluorescence yields were adjusted for baseline fluorescence using ASW + F/2. Fluorescence yield estimates from low light were used to derive the maximum PSII photochemical efficiency (*F*_v_/*F*_m_, where *F*_v_ = *F*_m_ − *F*_o_; dimensionless, with *F*_o_ = initial fluorescence and *F*_m_ = maximum fluorescence) using a model describing the light dependency of PSII photochemistry (as per [41]).

### 2.3. Symbiodiniaceae and Bacterial Cell Density Analysis

Cell counts were conducted through flow cytometry for each Symbiodiniaceae culture immediately before volatilomic, metabolomic, and elementomic samples were collected. Specifically, 1 mL of each culture was collected directly from culture flasks, from which an aliquot of 100 µL was collected, diluted 1:10, and directly used for flow cytometry analysis (CytoFLEX S, Beckman Coulter, Brea, CA, USA). Symbiodiniaceae cells were excited at 488 nm, identified according to their chlorophyll fluorescence (650 nm), and subsequently enumerated. Sample blanks (*n* = 6) were run alongside, and their average number of events was subtracted from each sample (blank correction). The Symbiodiniaceae flow cytometry gating strategy is shown in Appendix A.

### 2.4. Elementomics Data Collection and Analysis

Elementome data collection and processing followed the method described by [9]. Before sample collection, all plasticware and glassware were immersed in a Micro-90 solution (2% for 24 h), rinsed with ultra-pure water, followed by an acid-wash solution (10% hydrochloric acid for 24 h), before a final ultra-pure rinse and atmosphere drying [24]. All acids and reagents used were of analytical-grade purity and used without further purification. A 50 mL aliquot of each culture was collected into pre-cleaned falcon tubes under a class 100 laminar flow [42]. The samples were centrifuged (Rotanta 460R centrifuge, Hettich, Kirchlengern, Germany) at 5 min at the sample experimental temperature (e.g., 26 °C or 33 °C) at 1150× *g*. The supernatant was discarded, and 20 mL of sterile 0.1 M TRIS buffered saline solution was added to the centrifuge tubes, and algal pellets were resuspended before being re-centrifuged. This was repeated three times to remove any sorbed elements [9], and the final pellets were freeze-dried (Alpha 2–4 LDplus freeze dryer, CHRIST, Osterode am Harz, Germany) before being weighed and then sub-sampled for Carbon (C) and Nitrogen (N) analysis using a LECO TruMac Carbon Nitrogen Analyser (LECO Castle Hill, Castle Hill, NSW, Australia) and for the remaining elements using an Agilent Technologies 7700s-series inductively coupled plasma mass spectrometer (ICP-MS) (Mulgrave, Australia). For C and N analysis, manufacturer methods for soil and plant material when utilising a furnace temperature of 1200 °C were followed. A calibration standard (LECO) was used, and sample blanks and standards were utilised prior to sampling. The pellets for ICP-MS analysis were digested with a mixture of 100 μL of HNO_3_ (67–69% *w*/*w*, Choice Analytical, Thornleigh, NSW, Australia) and 100 μL of H_2_O_2_ (30–32% *w*/*w*, Seastar Chemicals, Sidney, BC, Canada) and incubated overnight. Samples were subsequently diluted to a final volume of 5 mL with ultra-pure water (18.2 MΩ; Merck Millipore, Burlington, MA, USA) and filtered using 0.2 μm syringe filters (Captiva Econoflters, Agilent Technologies, Santa Clara, CA, USA). High-purity ICP-MS standard calibration solutions for external calibration (Choice Analytical, Thornleigh, NSW, Australia) were diluted in aqueous solution of 3.3% HNO_3_ and 1.5% H_2_O_2_. Procedural blank samples (TRIS) were also run to check for potential contamination in the methodological process and came back negligible (<1%). The ICP-MS was equipped with a MicroMist nebuliser, Scott double-pass spray chamber (2 °C), s-lenses, a platinum sampler, and skimmer cones. Helium (4.2 mL min^−1^) was used as a collision gas to reduce polyatomic interference. A 100 ng mL^−1^ solution of rhodium in 1% HNO_3_ was used as an internal standard and delivered post-pump via a T-piece. ICP-MS parameters are detailed in Appendix A. Agilent ICP-MS Chemstation was used to acquire and process all data. Elementome data were normalised to the mass of the sample and to the Symbiodiniaceae cell density (Table 1).

### 2.5. Metabolomics Data Collection and Analysis

Metabolomics samples and analysis were conducted following the methods described in [43]. A 50 mL aliquot was taken from each culture, and the Symbiodiniaceae cells were concentrated by centrifugation at 700× *g* for 5 min at 26 °C (control) or 33 °C (heat stress), the media discarded, and Symbiodiniaceae pellets snap frozen in liquid nitrogen for metabolite profiling. All subsequent steps were performed at 4 °C to prevent metabolite losses during extraction. Metabolite extraction, analysis, and data processing are based on previous methods^60^. To first remove residual salts (which affect GC-MS analysis), each pellet was resuspended in 500 µL cold (4 °C) ultra-pure water, gently agitated for 10 s, centrifuged at 1500× *g* for 5 min at 4 °C, and the supernatant discarded.

Pellets were frozen at −80 °C for 1 h and lyophilized at −105 °C for 18 h. The semi-polar metabolites were extracted by adding approx. Ten mg acid-washed glass beads to each pellet and 200 µL 100% cold (−20 °C) methanol spiked with 20 µg/mL final concentration of the internal standard (IS) D-sorbitol-6-^13^C, and cells were lysed using a bead mill at 50 Hz for 3 min. A further 800 µL of 100% cold methanol (+IS) was added to each cell slurry, and samples were vortexed for exactly 1 min each. Cell debris was pelleted at 3000× *g* for 30 min at 4 °C, and the supernatant was transferred to a new 2 mL Eppendorf. To each cell debris, a further 1 mL 50% cold (−20 °C) methanol was added, and samples vortexed for 30 s. Cell debris was pelleted at 3000× *g* for 30 min at 4 °C, and the supernatant was combined with the 100% methanol extracts. Samples were centrifuged at 16,000× *g* for 15 min at 4 °C, and 5 × 50 µL (250 µL total volume) dried in a glass insert in a concentrator at 30 °C.

### 2.6. Online Derivatisation and Gas Chromatography-Mass Spectrometry Analysis

Dried samples for targeted analysis were derivatized online using the Shimadzu AOC6000 autosampler robot. Derivatization was achieved by adding 25 µL of methoxyamine hydrochloride (30 mg/mL in pyridine), followed by shaking at 37 °C for 2 h. Samples were then derivatized with 25 µL of *N*,*O*-*bis*(Trimethylsilyl)trifluoroacetamide (BSTFA) with 1% Trimethylchlorosilane (TMCS) (Thermo Scientific, Waltham, MA, USA) for 1 h at 37 °C. The sample was left for 1 h before 1 µL was injected onto the GC column using a hot needle technique. Split (1:10) injections were done for each sample. The GC-MS system used consisted of an AOC6000 autosampler, a 2030 Shimadzu gas chromatograph, and a TQ8040 quadrupole mass spectrometer (Shimadzu, Kyoto, Japan). The mass spectrometer was tuned according to the manufacturer’s recommendations using tris-(perfluorobutyl)-amine (CF43). GC-MS was performed on a 30 m Agilent DB-5 column with a 1 µm film thickness and a 0.25 mm internal diameter column. The injection temperature (inlet) was set at 280 °C, the MS transfer line was at 280 °C, and the ion source was adjusted to 200 °C. Helium was used as the carrier gas at a flow rate of 1 mL/min, and argon gas was used as the collision cell gas to generate the MRM product ion. The analysis of the derivatized samples was performed under the following temperature programme: start at injection 100 °C, hold for 4 min, followed by a 10 °C min^−1^ oven temperature ramp to 320 °C, and final hold off for 11 min. Approximately 520 quantitative multiple reaction monitoring (MRM) targets were collected using the Shimadzu Smart Database, along with a qualifier for each target, which covers about 350 endogenous metabolites and multiple ^13^C-labelled internal standards. Both chromatograms and MRMs were evaluated using the Shimadzu GCMS browser v3 and LabSolutions Insight software v3.8 SP1.

### 2.7. Metabolite Data Analysis

Metabolite data were normalised to peak area of the internal standard D-sorbitol-6-^13^C and then to total cell density of the 50 mL aliquot (Appendix A, Table 1). To test for overall differences in metabolite pools between culture treatments, statistical analyses were performed using MetaboAnalyst 5.0, where data were tested for normality and homogeneity and log-transformed where necessary. Data were then evaluated by Principal Component Analysis (PCA). Univariate (*t*-tests) tests were performed to identify individual metabolites that varied significantly between the treatment groups. Significant individual metabolites were determined based on a False Discovery Rate (FDR) corrected significance value (*p*_adj_ < 0.05).

### 2.8. Volatile Sample Collection

Biogenic volatile organic compounds (BVOCs) were sampled from live cultures of Symbiodiniaceae as per [44]. Sample aliquots of 50 mL were sealed in sterile crimp cap vials (100 mL) and maintained under matching growth conditions during sampling. Technical duplicates were collected from each biological replicate (*n* = 6). BVOCs were collected by purging each vial for 15 min with instrument-grade air at 100 mL/min (BOC Gases, Linde Group, North Ryde, NSW, Australia). The outlet of this purge passed through thermal desorption (TD) tubes (200 mg Tenax TA; Markes International Ltd., Llantrisant, UK). Immediately following sample purging, TD tubes were capped and stored at 4 °C until analysis. Water blanks (*n* = 6) consisting of culture medium (autoclaved and filter sterilised (0.22 µm) artificial seawater (ASW) and F/2 media) were also run to capture any background BVOCs.

### 2.9. GC×GC-TOFMS Analysis of BVOCs

Sorbent tubes were analysed using comprehensive two-dimensional gas chromatography coupled with time-of-flight mass spectrometry (GC×GC-TOFMS) (LECO, Castle Hill, NSW, Australia). Prior to desorption, an internal standard was added to each TD tube (0.2 μL of 10 ppm chlorobenzene-d5 (HPLC grade, Sigma-Aldrich, Castle Hill, NSW, Australia) in methanol (HPLC Grade, Chem-Supply PTY Ltd., Port Adelaide, SA, Australia) using an eVol^®^ XR handheld automated analytical syringe (SGE Analytical Science, Weatherill Park, NSW, Australia) to enable peak normalisation. A Markes Unity 2 Thermal Desorber and Series 2 ULTRA multi-tube autosampler (Markes International Ltd.) were used to thermally desorb the samples before injection onto the Pegasus 4D GC×GC-TOFMS. During thermal desorption, tubes were desorbed at 300 °C for 2.5 min, then focused onto the cold trap held at −30 °C before quickly being heated to 300 °C to push the sample onto the primary GC column via a transfer line. The GC×GC-TOFMS was set up with a Rxi-624Sil MS column (30 m, 0.25 mm inner diameter, 1.4 μm film thickness, Restek Corporation, Bellefonte, PA, USA) in the first dimension and a Stabilwax column (2 m, 0.25 mm inner diameter, 0.5 μm film thickness, Restek Corporation) in the second dimension. Helium (high purity, BOC, Sydney, NSW, Australia) was used as the carrier gas with a flow rate of 1 mL/min. The temperature programme was as follows: initial temperature of 35 °C with a 5 min hold, followed by an increase to 240 °C at a rate of 5 °C/min with a 5 min hold. Temperature offsets of +5 °C and +15 °C for the modulator and second dimension column, respectively, were used. The modulation period was 5 s and included a 1 s hot pulse. The TOFMS collected data at a rate of 100 spectra/s with a mass range of 29–450 amu.

### 2.10. BVOC Data Pre-Processing

Data processing was performed using ChromaTOF^®^ (version 4.51.6.0; LECO). A signal-to-noise ratio of 150 was used with a baseline offset of 0.8. The peak widths for the first and second dimensions were 30 s and 0.15 s, respectively. Analyte identification was carried out using the National Institute of Standards and Technology (NIST) Mass Spectral Library, where a match threshold of 80% was required. Analytes were aligned using the statistical compare tool within the software, where a spectral match threshold of 60% was required to match analytes across samples. After alignment, the analytes were normalised against the internal standard. Known artefacts and environmental contaminants were excluded, and a final compound table was created. Blank subtraction was subsequently done using the blank water samples (*n* = 5), whereby the analyte needed to be present in an abundance of more than 50% of the blank water samples to be retained. In addition, analytes that were not present in more than three of the biological replicates per sample type were removed. The resulting analytes were then normalised against the cell count (cell/mL; Appendix A). Compounds were grouped into their respective compound classes to investigate volatilome variability across the treatment and control groups.

### 2.11. Integration Analysis and Statistics

To integrate the independent omic datasets, the normalised relative abundance of elements, metabolites, and BVOCs for each sample was combined into a single dataset. Log transformed and mean-centred data were then evaluated by Principal Component Analysis (PCA) in R (v4.3.0) using the packages *ggforce*, *ggfortify*, *ggnewscale*, *mice*, *ggalt*, and *dplyr*. A PERMANOVA was performed (PRIMER v4) to test for overall differences across the integrated dataset between culture treatments and univariate (*t*-tests, Mann–Whit) analysis to identify individual compounds/elements that varied significantly between the treatment groups. Significant individual compounds/elements were determined based on a False Discovery Rate (FDR) corrected significance value (*p*_adj_ < 0.05).

### 2.12. Correlation and Network Analysis

Spearman Rank Correlation with Holme *p*-value adjustments was performed on the compounds/elements that were significantly different between control and treatment groups (*t*-test and Wilcoxon rank-sum test, *p*_adj_ < 0.05) using the Psych package (version 2.3.3) in R (version 1.3.1093; R Core Team, 2023), as per [45]. An additional network analysis was performed using this data in Cytoscape 3.9.1. [46] to visualise the correlations between elements, metabolites, and volatiles. Compounds were arranged into clusters where each compound is correlated to at least three other compounds within the cluster, as per [45].

## 3. Results

### 3.1. Symbiodiniaceae Photophysiology

The maximum quantum yield of PSII (*F*_v_/*F*_m_) of Symbiodiniaceae cultures under heat stress declined over time and was significantly lower than controls on days 8 and 9 (Appendix A, Figure 1). Cell densities did not differ between control (150,067 ± 18,076 cells mL^−1^) and treatment (126,700 ± 6283 cells mL^−1^) cultures at the end of the experiment (Welch’s two-sample *t*-test, t = 1.221, df = 6.191, *p*-value = 0.267).

### 3.2. Chemical Omics Approaches in Isolation

Elementomics was used to retrieve the concentrations and stoichiometry of 17 elements: seven macronutrients and ten micronutrients (Appendix A). From the three applied methods, the principal component analysis for the elementomics data had the greatest overlap between treatments (Figure 2A). Despite the overlap, differences were detectable between the control and heat stress treatments. Collectively, the first (PC1) and second (PC2) principal components explained 85.48% of the variance in the data. PC1 accounted for 69.73% of variance, with higher calcium (Ca), strontium (Sr), and potassium (K) in the control samples driving most separation along this axis. Potassium was the only element significantly different between the control and heat stress groups when elements were considered on an individual basis (t_(2)_ = 4.06, *p*_adj_ = 0.04). PC2 explained 15.75% of the variance with a large spread within the heat stress treatment on this axis, resulting from the large difference in carbon (C) and nitrogen (N) values for the heat stress samples. The largest and smallest C and N values from the study were from the heat stress treatments. Both the controls and heat stress had a C:N:P ratio below the Redfield ratio (106:16:1; [47]), with values of 14:3:1 and 24:6:1, respectively.

Application of metabolomics resulted in a total of 120 metabolites being detected across both treatments (Appendix A). Principal component analysis revealed separation between the groups on PC1, which represented a 36.62% separation between the treatment groups (Figure 2B). The separation is driven by increases in glutathione, xanthine, guanine, and tryptophan and decreases in glucose, citric acid, glutamic acid, aminobutyric acid, and galacturonic acid in the heat-stressed cultures. Of the metabolites, 24 differed significantly under heat stress (Wilcoxon rank-sum test, *p*_adj_ < 0.05, Appendix A). Of these, 15 metabolites were higher in relative abundance in heat stress cultures, including inositol, indole-3-acetic acid, and tryptophan, and 9 metabolites were lower in relative abundance in heat stress cultures, including the TCA intermediates citric acid, aconitic acid, and itaconic acid.

Volatilomics analysis resulted in a total of 163 volatile organic compounds detected and identified post-filtering (Appendix A), 66 of which were found in the heat stress only, 11 solely in the controls, and 86 were common amongst the two treatments (Figure 3). A difference in chemical composition was also evident through the PCA, whereby a separation was observed in the PCA along PC1 (38.24%; Figure 2C) with the majority of compounds associated with treatment conditions. Heat stress cultures had lower variability between replicates, as demonstrated by the tighter clustering (Figure 2C). Of the volatile metabolites found in both treatments, 27 were significantly more abundant in heat stress cultures (Wilcoxon rank-sum test, *p*_adj_ < 0.05, Appendix A), including ozone-depleting dibromomethane and 2-ethyl-1,3-dimethylbenzene, which had previously been detected in heat-stressed Symbiodiniaceae [44]. Only benzonitrile and 1,3-bis(1-methylethenyl)-benzene were less abundant in heat stress (Wilcoxon rank-sum test, *p*_adj_ < 0.05).

Six carboxylic acids were found exclusively in the heat stress samples, including acetic acid, butanoic acid, octanoic acid, and propanoic acid (Appendix A). Two out of the seven halogenated compounds detected were only seen in the heat stress treatment; the remaining five were found in both. A number of hydrocarbons were detected in the heat stress treatment only, particularly for alkenes, where 60% of the compounds were in the heat stress treatment only (Figure 3). Sulphur-containing compounds are often reported in the volatilome of marine species; a number of these were present in both sample types; however, none were found in the control samples, and only two were found in the treatment samples. Dimethyl sulphide (DMS) was detected in both sample types; however, the signal in the controls was found to be ~2 times higher than that of the treatment samples.

### 3.3. Integration of Chemical Omics Approaches

Integration of elementomics, metabolomics, and volatilomics resulted in 53 analytes being significantly different between the control and heat stress cultures (Figure 4; Appendix A). Of these, there was one element (potassium, K), 22 metabolites, and 30 BVOCs (Appendix A). A total of 12 analytes were significantly more abundant in control cultures (including K, 9 metabolites, and 2 BVOCs), while the remaining 41 were more abundant in heat stress cultures, comprising 15 metabolites and 26 BVOCs. PCA analysis of the integrated data further supports a separation between the treatments, driving 37.08% of the data separation (Figure 2D).

### 3.4. Correlation and Network Analysis

Spearman rank correlation performed on only compounds that differed significantly between control and treatment groups resulted in 89 positive correlations and 1 negative correlation (Appendix A; Figure 5). When arranged into clusters where each compound is correlated to at least three other compounds within the cluster, five independent correlation networks were produced. The largest correlation network included 23 compounds, consisting of two metabolites and 21 volatiles. Of these compounds, six BVOCs (2-cyclopropyl-butane; 2-butyltetrahydrolfuran; 2-methyl-pentane; tetrahydro-2-furanmethanol; 3-methyl-2-hexene; and anisole) were correlated to >8 other compounds in the cluster. The remaining clusters were comprised of metabolites only, one of which included TCA cycle intermediates (citric, itaconic, and aconitic acids), another with all four DNA bases and the glutamate precursor 5-oxoproline, and then two networks comprising two metabolites, lactose with glycerol-3-phosphate, both involved in the glycolysis pathway, and biochemically distinct inositol with phenylalanine.

## 4. Discussion

The integration of omics data holds significant promise for advancing our understanding of complex symbiotic relationships and organisms with unresolved or non-model biochemistry. In this study, independent analyses unveiled and supported prior evidence of the crucial pathways associated with thermal stress responses in Symbiodiniaceae. However, the strength of the integrated approach lies in its ability to bridge gaps in metabolic pathways, especially for analytes with unknown functions and those lacking model organisms, as exemplified by the Symbiodiniaceae.

Individually, each chemical omic approach distinguished, to varying degrees, specific analytes that shift under thermal stress. Volatilomics data revealed the largest number of significantly altered analytes, with 21 significant BVOCs; this is likely due to the chemical diversity of BVOCs, although the function of most of these analytes remains unresolved [44]. Many (53%) of the volatiles detected were found to be shared across the treatment and control conditions, while 40% were unique to the heat stress cultures. This suggests heat stress causes the release of additional volatiles that are not emitted or are below detection levels under stable conditions. Indeed, an increase in the diversity of volatiles was described previously in *D. trenchii* under heat stress [44] and was suggested to be due to the synthesis of specific compounds in response to heat stress in this relatively heat-tolerant Symbiodiniaceae species, as observed in higher plants [48]. Two out of the seven halogenated compounds detected were unique to the heat-stressed cultures and have been found in association with marine environments, such as the ozone-depleting compound dibromomethane [49,50]. A number of hydrocarbons were also detected in the heat stress cultures only, including cyclohexanone, which was previously found to be emitted from corals [51] and associated with thermal tolerance in Symbiodiniaceae [44]. Two sulphur-containing compounds (aminomethanesulfonic acid and carbon disulfide) were only detected in heat-stressed cultures, and thiirane was significantly more abundant under heat stress (Appendix A). Sulphur-containing volatiles have been previously suggested to be involved in stress response in Symbiodiniaceae and trophic interactions in coral reefs [44]. While previous studies detected a decrease in the emission of volatile sulphur compounds under heat stress by *D. trenchii* [52], the increased diversity of ulfur-containing compounds and the significant increase in the emission of thiirane under heat stress continue to suggest a role for volatile sulphur compounds in the Symbiodiniceae stress response.

The metabolomics data highlighted the next highest number of changes in the relative abundance of specific metabolites in response to thermal stress. Indole-3-acetic acid (IAA) and tryptophan increased under heat stress; these metabolites have recently been identified in the molecular interplay between Symbiodiniaceae and bacteria, with IAA production from bacteria enhancing Symbiodiniaceae growth, and the release of tryptophan by Symbiodiniaceae could support IAA biosynthesis [43]. Indeed, the Symbiodiniaceae bacteriome has been shown to shift under heat stress [53]. Therefore, the modifications in the relative abundance of these metabolites might be due to a shift in the microbiome composition and Symbiodiniaceae-bacteria interactions.

Another metabolite highlighted as significantly increasing in relative abundance in the thermal stress cultures was inositol, a response that has consistently manifested in the metabolome of Symbiodiniaceae under heat stress [16,54,55,56,57]. Inositol is a sugar alcohol that naturally occurs in different stereoisomers (e.g., myo-, scyllo-, chiro-), which have roles in cellular recognition, signalling, development [58,59] and nutrient cycling [60]. *Myo*-inositol is also a known osmolyte in plants, some macro-algae species, and a handful of free-living micro-algae species, including Symbiodiniaceae [61,62]. This inositol stereoisomer is suspected to protect cellular structures from reactive oxidizers [63]. Under high temperatures, Symbiodiniaceae have been observed to increase production and accumulation of reactive oxygen species (ROS) due to heat-induced damage to the photosynthetic apparatus [64], thus this profile change is further evidence for the role of inositol as a cellular protectant from stress in Symbiodiniaceae.

In the elementomic data, potassium was the only element that significantly decreased in relative abundance in response to heat stress. High temperatures can disrupt the ion balance and membrane permeability within cells. As potassium is an essential cation involved in various cellular processes, including enzyme activation, osmoregulation, and membrane potential, heat stress may lead to an imbalance of ions, causing a leakage of potassium ions from the cells, facilitated by increased membrane permeability [65]. In addition, heat stress and resulting oxidative stress may inhibit the function of enzymes transporting potassium across cell membranes, leading to reduced uptake or increased efflux of potassium. In plants, ROS production from both photosynthetic electron transport and NADPH-oxidising enzyme reactions has resulted in membrane damage in potassium-deficient plants, suggesting potassium is a key element to alleviate stress [66]. The relatively fewer significant changes in element relative abundance is not unexpected, as the total elemental composition and relative abundance of each element are not necessarily changing under heat stress, but the elements may be repurposed into different chemicals and analytes, as demonstrated in the metabolomics and volatilomics data. Further, a strength of elementomics is to consider the amount and stoichiometry of elements to detect subtle changes in biogeochemical requirements that are lost when elements are analysed on an element-by-element basis [67]. The PCA highlights that under heat stress, Symbiodiniaceae have unique elemental requirements, in support of work from a previous Symbiodiniaceae study [9]. A strength of elementomics is therefore the ability to detect how fundamental resource requirements may change under future environmental conditions and better predict the resource needs of organisms.

Integrating the analytes with a correlation analysis revealed several links between compounds that suggest metabolic products may have been reutilized in response to potential oxidative stress (Figure 6). Citric acid, aconitic acid, and itaconic acid, constituents and end products involved in the tricarboxylic acid (TCA) cycle, were positively correlated to each other and all found to decrease under heat stress. A decrease in TCA cycle-related compounds may suggest accelerated respiration to generate energy needed for the synthesis of antioxidant compounds under heat (and consequently oxidative) stress [68]. Further evidence of oxidative stress can be found in the correlation between sucrose and tetrahydro2-furanmethanol and their increase in abundance under heat stress. Sucrose, a primary photosynthetic end product, accumulates in higher plants under stress and scavenges oxygen free radicals [69]. Furfuryl alcohol (2-furanmethanol) is a by-product of furfural degradation—a process that converts furfural to 2-oxoglutarate for use in the TCA cycle—and is similarly a known ROS scavenger [70]. Of interest, 2-furanmethanol was also significantly correlated with the metabolite glyceric acid, which was separately correlated with the metabolites cysteine and threonic acid. In central metabolism, glycerol is phosphorylated to glycerol-3-phosphate, which is oxidised to dihydroxyacetone phosphate and subsequently enzymatically converted to glyceric acid. Although glycerol-3-phosphate was not identified as correlated with these compounds (Figure 5), it was significantly more abundant in heat-stressed cultures, while glyceric acid was significantly reduced. In glycolysis, glyceric acid can be further metabolised to produce energy or be used for the synthesis of other metabolites. Thermal and oxidative stress in Symbiodiniaceae have been shown to result in changes in central energy processes such as glycolysis [56], and while there is no known metabolic pathway linking glyceric acid to 2-furanmethanol, they are similarly indicating changes in central metabolism and oxidative responses.

The integration of datasets as presented here could play a pivotal role in enhancing our understanding of the functional role of BVOCs in Symbiodiniaceae heat stress responses. Glyceric acid and 2-furanmethanol were significantly correlated with three other BVOCs—dibromomethane, thiirane, and anisole—and glyceric acid was independently correlated with cysteine and threonic acid. All three metabolites were significantly lower in heat stress cultures, while the BVOCs were significantly higher. Thus, the correlation between these compounds could suggest an indirect link between these volatiles and increased central metabolism and glycolysis activity, oxidative stress responses, or another metabolic heat stress response, either by the Symbiodiniaceae or associated microbes. For example, dibromomethane, a simple halogenated hydrocarbon, has been previously detected in marine algae volatile emissions [49] and marine microbiomes [50]. Thiirane (ethylene sulphide) is a small sulphur-containing ring compound that, while not commonly encountered in biological systems as a natural product or metabolite, is a highly reactive substance, particularly with sulphur nucleophiles, and can be oxidised under thermal conditions to ethylene episulfoxide. This is perhaps further evidence for the apparent role of sulphur-containing analytes in Symbiodiniaceae heat stress. Anisole (methoxybenzene) is a simple aromatic compound that can be found naturally in some plants but also as a marine pollutant [49,71]. These findings present potential candidates for more in-depth investigations, contributing to a broader understanding of metabolic processes in complex symbioses.

Due to the few significant changes in elements, the elementomic data were limited in their capacity to strengthen the interpretation of biochemical interactions when correlated with other omics approaches. For example, while potassium was identified as significant in the elementomics analysis, it was not found to be correlated to any other analytes. But when the overall profiles are considered in concert, we continue to find evidence of a redirection of metabolic pathways to cope with stress conditions. For example, broader-scale heat stress and physiological and metabolic responses in Symbiodiniaceae may alter the demand for potassium, leading to its redistribution within the cell or its extrusion from the cell. In addition, the largest and smallest C and N values from the study were from the heat stress treatments, and both treatments had a C:N:P ratio below the Redfield ratio (106:16:1; [47]), with values of 14:3:1 and 24:6:1, respectively. Previous work has found that Symbiodiniaceae typically have a lower C:N:P ratio than the Redfield ratio [9]. The difference in C:N:P ratio between the control and heat stress treatments suggests changes in nutrient assimilation, utilisation, and allocation under heat stress. The shift in the C:N:P ratio could indicate a reallocation to prioritise stress response mechanisms, such as heat shock proteins or antioxidant production, at the expense of normal energy and growth-related processes in heat-stressed Symbiodiniaceae cells. Indeed, the observed lower abundance of TCA cycle intermediates further supports a shift from energy-producing pathways to stress responses.

As omics analyses become increasingly feasible and affordable, and with the continuous growth of technology and databases, the prospects for integrated omics studies in the fields of Symbiodiniaceae and coral biology are expected to expand. This trajectory aligns with the field’s recognition of the importance of such applications, as underscored by the priority attributed to them in recent discussions [27,37]. As we embark on this trajectory, there are considerations in data handling that we would like to highlight. Firstly, normalisations must be carried out in omics analyses to account for variations during sample preparation and instrument functionality. Here, we describe the stepwise normalisations for each individual omics approach to a per-cell relative abundance to achieve a cross-omic comparable unit. Elementomic, volatile, and metabolite data were first normalised to an internal standard concentration to account for instrument variability, as is recommended for mass spectral data [72]. Normalising volatilomic data to a standardised unit measure of sample biomass (e.g., surface area or cell density) is typical [44,52]. While metabolomic data is better normalised to the biomass of the sample extracted from [43,73], the cell density of the original sample is also commonplace for [74]. For elementomics, concentration data (µg kg^−1^) was normalised to dried sample biomass. To account for differences in cell size, this can be further divided by the number of cells in that biomass. While multiple normalisation steps are necessary to align across omics platforms, they can distort the data and thus should be carefully considered and described in detail in experimental reports.

As data availability and pipelines continue to improve, it provides exciting possibilities to enhance the outcomes of omic integration experimental designs. For example, the integration of data from single-cell omics holds tremendous promise. Profiling *in hospite* cells at individual positions within the coral tissue can significantly contribute to our understanding of the spatial distribution and ecological roles of Symbiodiniaceae in the coral holobiont [75]. Emerging techniques such as single-cell metabolomics [76] could be seamlessly integrated with single-cell transcriptomics [16,77] and potentially with single-cell elementomics (e.g., [78]), collectively shedding light on biogeochemical patterns and unveiling genetic, metabolic, or elemental biomarkers. While the application of single-cell volatilomics is challenging due to the detection threshold for volatile analysis, there remain valuable avenues for data collection. Integrating volatile data obtained from larger specimens or cultures with known stress-related gene or metabolite expressions from single cell analyses could provide insights into interspecies interactions and potentially contribute to the development of non-invasive diagnostic tools. This multidimensional approach to omics integration, incorporating both single-cell techniques and cross-disciplinary data synthesis, holds great promise for unravelling the intricacies of Symbiodiniaceae and coral biology in greater detail.

This study offers crucial insights into how Symbiodiniaceae metabolically respond to thermal stress (Figure 6). It reveals compelling evidence of analyte reallocation, indicating a strategic shift to prioritise stress response mechanisms over typical energy and growth-related processes. Notably, the metabolomic and volatilomic analyses reveal alterations in TCA cycle-related compounds and an augmented presence of antioxidant compounds. This suggests an accelerated respiratory response and heightened oxidative stress reactions, findings that align with the evidence of decreased photosynthetic efficiency. Integrating multiple omics approaches can therefore provide a powerful tool for examining the physiological responses of marine organisms to environmental stress, including corals.

## Figures and Tables

**Figure 1 microorganisms-12-00317-f001:**
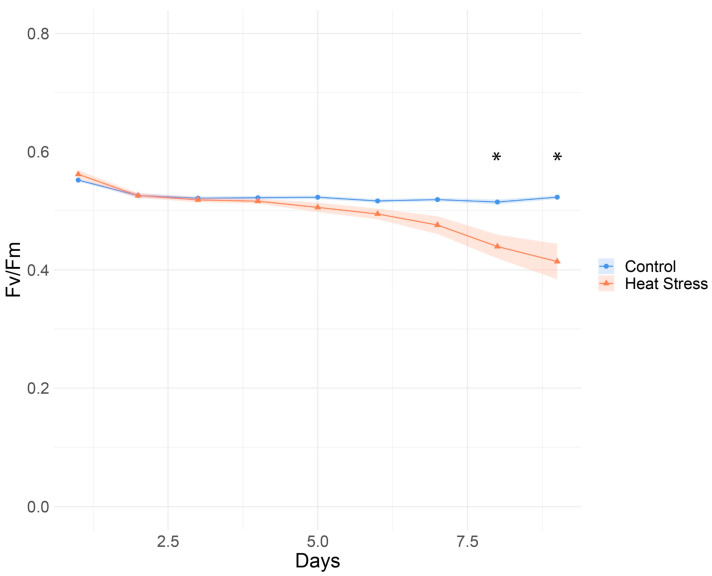
Photophysiological parameters of *Durusdinium trenchii* at control (26 °C) and heat stress (33 °C) temperatures. Quantum yield (*F*_v_/*F*_m_) was measured daily for heat-stressed and control cultures following dark acclimation. Points are mean, with shading indicating the standard error. Asterisk indicates time points with significant differences in *F*_v_/*F*_m_ between the control and heat stressed cultures (Two tail *t*-test, *p*_adj_ < 0.05).

**Figure 2 microorganisms-12-00317-f002:**
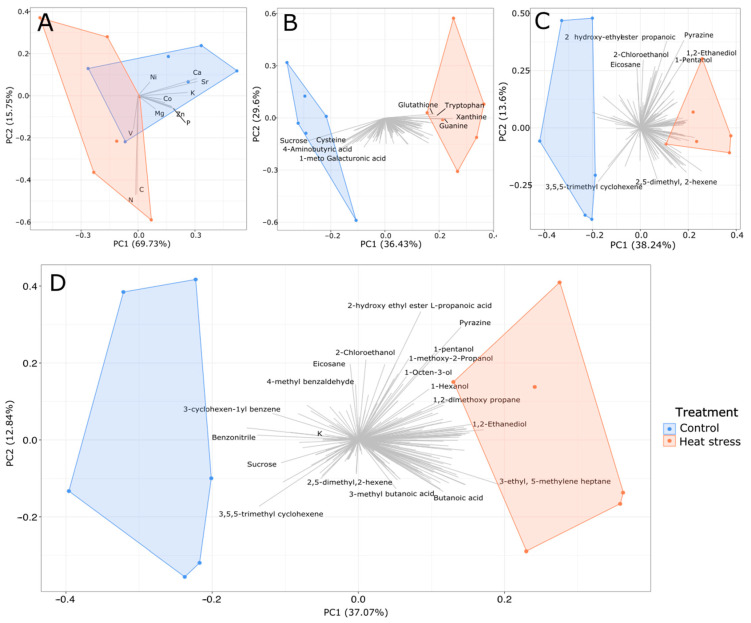
Principal Component Analysis of independent and integrated profiles of analytes extracted from *Durusdinium trenchii* at control (26 °C) and heat stress (33 °C) temperatures. Independent PCA of elementomic (**A**), metabolomic (**B**), and volatilomic (**C**) analyses, and the PCA for the integrated data set (**D**).

**Figure 3 microorganisms-12-00317-f003:**
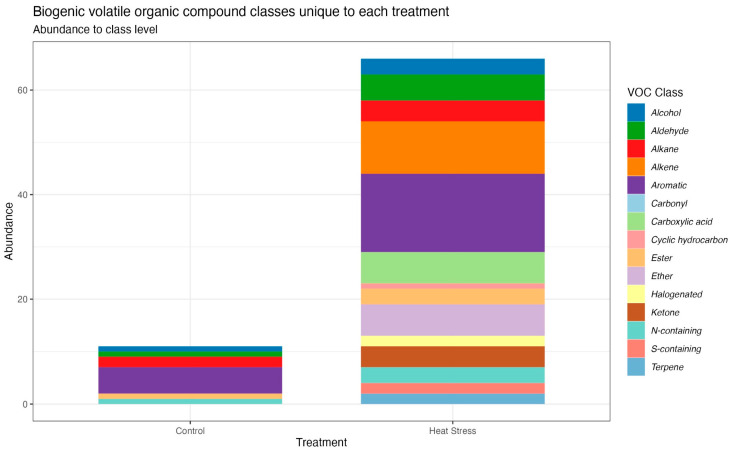
Unique biogenic volatile organic compound classes from *Durusdinium trenchii* at control (26 °C) and heat stress (33 °C) temperatures. Data represents the abundance of unique volatile compounds in each treatment after volatiles were identified in the library and assigned to a chemical class.

**Figure 4 microorganisms-12-00317-f004:**
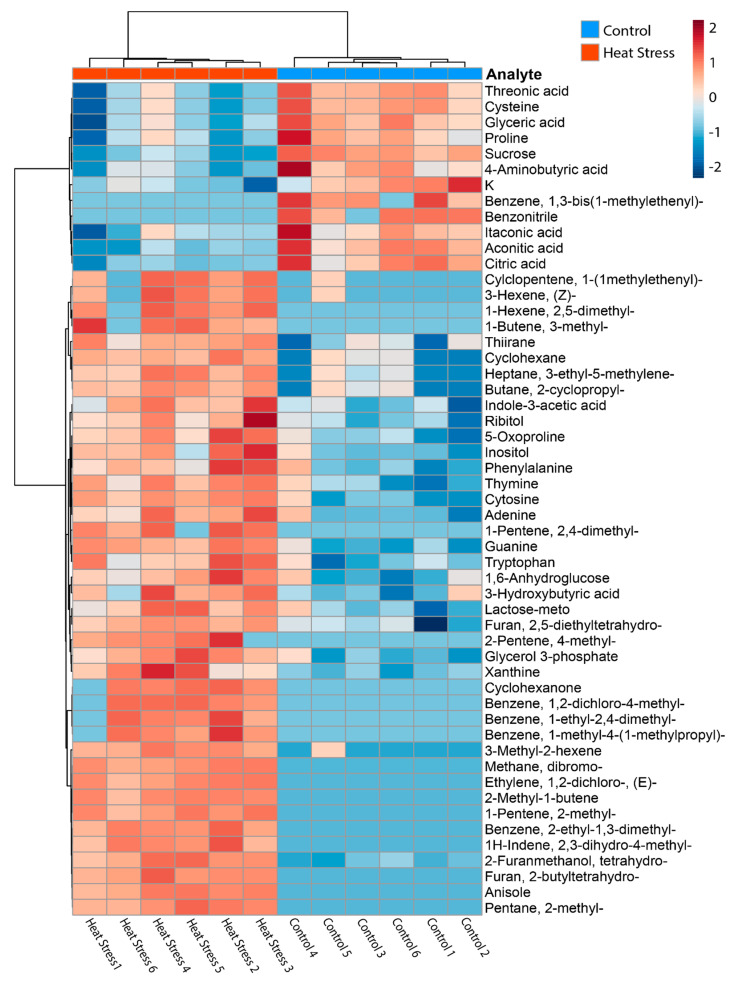
Heatmap of analytes identified as significantly different between control and heat stress cultures of *Durusdinium trenchii*. Only analytes detected as significantly different between the treatments by *t*-test are shown. Samples are clustered by Euclidean distance measure, and the relative abundance of each analyte is autoscaled.

**Figure 5 microorganisms-12-00317-f005:**
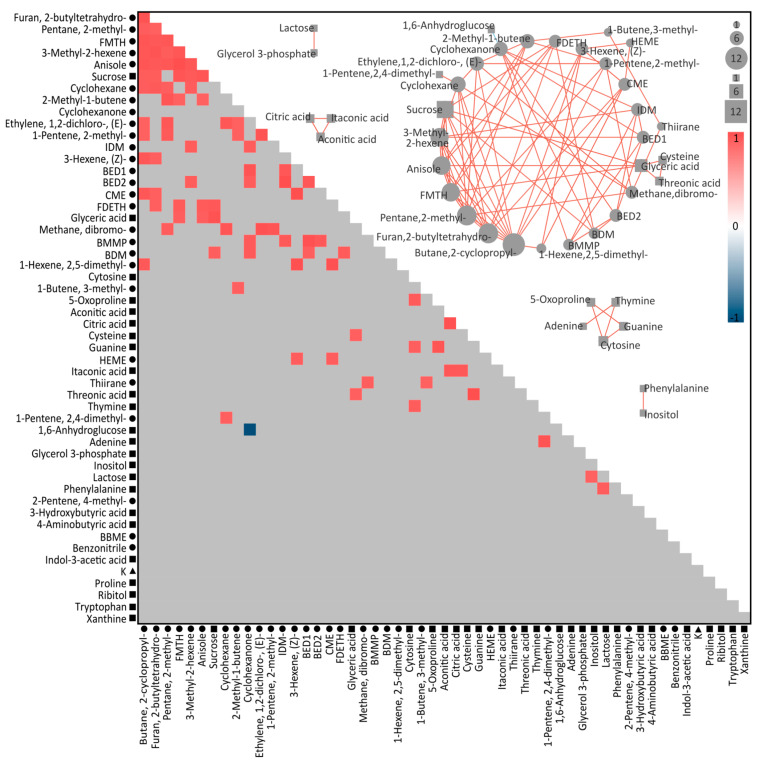
Heatmap displaying correlation coefficients between compounds (volatiles = circles, metabolites = squares, elements = triangles), produced by Spearman Rank correlation. Coloured cells indicate a Holme-corrected *p*-value of <0.05; red indicates a positive correlation, while blue indicates a negative correlation. Grey shading indicates non-significance. The networks displayed here depict correlations between compounds, with all compounds in the central network correlated to at least three other compounds. Red lines indicate a positive correlation, while blue indicates a negative correlation (Holme-corrected *p*-value < 0.05). The size of each node also indicates how many correlations a compound has, ranging from 1 to 12. Some compound names have been shortened for clarity: BDM = 1,2-dichloro-4-methyl-benzene; BBME = 1,3-bis(1-methylethenyl)-benzene; BED 1 = 1-ethyl-2,4-dimethyl-benzene; BED2 = 2-ethyl-1,3-dimethyl-benzene; BMMP = 1-methyl-4-(1-methylpropyl)-benzene; CME = 1-(1-methylethyl)-cyclopentene; FDETH = 2,5-diethyltetrahydro-furan; FMTH = tetrahydro-2-furanmethanol; HEME = 3-ethyl-5-methylene-heptaneand; and IDM = 2,3-dihydro-4-methyl-1H-indene.

**Figure 6 microorganisms-12-00317-f006:**
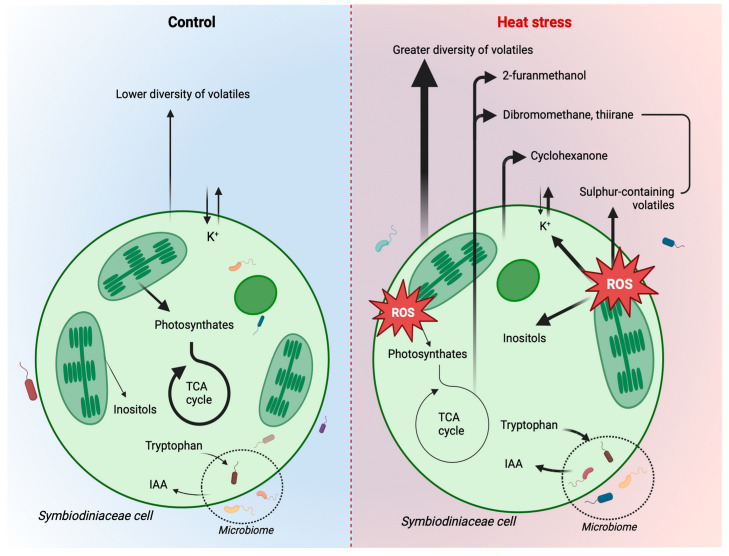
Conceptual diagram depicting the potential analyte patterns and interactions in Symbiodiniaceae under heat stress. Arrows demonstrate the analyte evidence from individual and integrated analytical approaches in a Symbiodiniaceae cell under control (**left**) and heat stress (**right**), with thicker arrows indicating increased relative abundance. ROS = reactive oxygen species.

**Table 1 microorganisms-12-00317-t001:** Description of initial data processing steps for each analytical method used in this study.

Method	Measurement	Outlier Consideration	Normalisation	Scaling
Elementomics	Fundamental suite of elements that make up the organism.		To mass and volume of culture and number of cells in the culture.	Log transformed and mean-centred
Metabolomics	The substrates, intermediates, and end products of cellular metabolism.		To internal standard relative abundance and cell debris pellet following extraction	Log transformed and mean-centred
Volatolomics	Biogenic volatile organic compounds.	Known contaminants and unknown analytes were removed. Blank subtraction and analyte filtering.	To internal standard relative abundance and number of cells in the culture.	Log transformed and mean-centred

## Data Availability

All data supporting our findings have been provided in the Appendix A.

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
