# Peer review of "Multi-Chemical Omics Analysis of the Symbiodiniaceae Durusdinium trenchii under Heat Stress"

_microorganisms, 2024, doi:10.3390/microorganisms12020317_

Round 1
Reviewer 1 Report
Comments and Suggestions for Authors
This manuscript entitled "Multi-chemical omics analysis of the Symbiodiniaceae Durusdinium trenchii under heat-stress" is an original article that investigated the endosymbiotic dinoflagellate Durusdinium trenchii, which is critical phytoplankton in the coral environment in Australia, by studying the heat stress at 33 ËšC and control 26ËšC and fill the scientific gap about the integrated chemical omics approach, combining elementomics, metabolomics, and volatilomics with a concern on the oxidative stress pathways to reveal the long term changes in the D. trenchii metabolism which may results in environmental crisis in the future by the global warming issues.
The introduction is well-written with an introduction including all the relevant items and information related to the symbiotic algae Symbiodiniaceae and its effect on the coral reef, ecosystem, and environmental changes. The authors incorporated pertinent citations in the research section, enhancing the manuscript's substance and contributing to the current knowledge in the scientific discipline of Symbiodiniaceae and related bioinformatics studies. The manuscript contained 75 references, comprising approximately 37 recent research studies published over the past five years. Among these references, eight were published in 2023, and no significant self-citations were found.
A total of one table (+ 4 supplementary) and four figures (+ 1 supplementary) represent the results and objectives of this study. But:
Figure 2. For PCA analysis, it will be clearer if the figure includes only the labeled arrows and removes the non-significant compounds.
Figure 5. The caption is not in the same line, also (volatiles = circle, metabolites = circle, elements = square). How do you differentiate between volatiles and metabolites while both are circles? Then you used, " Squares represent metabolites, and circles represent volatiles." Could you make the same symbol? If you used a square for elements, it should be in all figures as a square.
It is better to add a focused and specific conclusion about the effect of temperature change on the Symbiodiniaceae and coral reef at the end of the manuscript discussion rather than just address bioinformatics problems in data processing.
Corrections:
103: Durusdinium trenchii should be D. trenchii
154: Use g instead of RPM
486-501: The text is in a different format!
592: Acropora palmata should be italic
605: the title is in CAPITAL letters, it should be fixed as other reference format
607: Pocillopora eydouxi should be italic
631: Symbiodinium kawagutii should be italic
687: reference number 47 without publication year
The final decision is to accept this manuscript after minor revision.
Reviewer 2 Report
Comments and Suggestions for Authors
The authors provide new high-throughput methods to study the physiological responses of coralline Symbiodiniaceae Durusdinium trenchii under high-temperature treatments. I suggest the authors refer to the following comments and make revisions before this article can be accepted for publication in this journal.
General comments
Please summarize all the results in one figure so readers can fully understand the research results.
Specific comments
1. Line 106. Please provide the formula for artificial seawater (or reference)
2. Line 133. Please explain what Fm and Fo are.
3. Line 141. Please provide the wavelength of excitation you used in flow cytometry analysis.
4. Line 154. Please convert 3500 RPM to??? Xg.
5. Line 155. 0.1N TRIS buffered saline? Do you mean 0.1M? and please provide the pH value of this buffer.
6. Line 161. Please provide the full name of ICP-MS.
7. Line 187. “…on the methods described in60.” What is “in60”?
8. Line 188. What is the 500ul cold “MilliQ”? Do you mean the pure water generated by the MilliQ?
9. Line 194. What is “+IS”?
10. Line 206. BSTFA with 1% TMCS. What are BSTFA and TMCS?
11. Line 216. What are MRM and TMS?
12. Line 313. “…(Fig. 1A)”. It should be the Fig 2A.
Round 2
Reviewer 2 Report
Comments and Suggestions for Authors
I have no question and recommend this manuscript to be published in this journal.